# Multi-Camera Calibration Using Far-Range Dual-LED Wand and Near-Range Chessboard Fused in Bundle Adjustment

**DOI:** 10.3390/s24237416

**Published:** 2024-11-21

**Authors:** Prayook Jatesiktat, Guan Ming Lim, Wei Tech Ang

**Affiliations:** 1Rehabilitation Research Institute of Singapore, Nanyang Technological University, Singapore 308232, Singapore; guanming001@e.ntu.edu.sg (G.M.L.); wtang@ntu.edu.sg (W.T.A.); 2School of Mechanical and Aerospace Engineering, Nanyang Technological University, Singapore 637460, Singapore

**Keywords:** multi-camera calibration, active wand, triangulation, bundle adjustment

## Abstract

This paper presents a calibration approach for multiple synchronized global-shutter RGB cameras surrounding a large capture volume for 3D application. The calibration approach uses an active wand with two LED-embedded markers waved manually within the target capture volume. Data from the waving wand are combined with chessboard images taken at close range during each camera’s intrinsic calibration, optimizing camera parameters via our proposed bundle adjustment method. These additional constraints from the chessboard are developed to overcome an overfitting issue of wand-based calibration discovered by benchmarking its 3D triangulation accuracy in an independent record against a ground-truth trajectory and not on the record used for calibration itself. Addressing this overfitting issue in bundle adjustment leads to significant improvements in both 3D accuracy and result consistency. As a by-product of this development, a new benchmarking workflow and our calibration dataset that reflects realistic 3D accuracy are proposed and made publicly available to allow for fair comparisons of various calibration methods in the future. Additionally, our experiment highlights a significant benefit of a ray distance-based (RDB) triangulation formula over the popular direct linear transformation (DLT) method.

## 1. Introduction

Multi-camera systems are essential in various computer vision applications, including motion capture studios [1] and 3D pose estimation [2], where the capture volume is situated at the center and surrounded by cameras within a large volume. These applications require an accurate calibration process to estimate the intrinsic and extrinsic parameters of each camera. Intrinsic parameters, such as focal length, optical center, and distortion parameters, are essential for understanding the internal characteristics of each camera. Extrinsic parameters, which define the position and orientation of each camera in a common reference frame, enable the triangulation of 3D object positions from multiple camera perspectives. Although calibration can be time-consuming and must be repeated after camera displacement, it is essential for maintaining multi-camera system accuracy and reliability.

Various camera calibration methods have been developed since the advent of digital cameras. No single approach suits all scenarios, as the optimal choice depends on the specific situation. Nonetheless, the most widely adopted method uses planar patterns with known geometry, such as a square grid, circle grid, chessboard, or fiducial marker. This method is easy to implement and effective for the intrinsic calibration of a single camera or the extrinsic calibration of stereo cameras observing the pattern simultaneously.

However, using planar patterns for the extrinsic calibration of multi-camera systems presents multiple challenges, particularly in large capture volumes with diverse viewpoints and large distances between cameras. It is important to ensure that the calibration points are well distributed across the entire image frame for more reliable estimation [3]. The patterns also need to be sufficiently large to compensate for their distance from the cameras, and the pattern should not move too quickly to prevent blurring of the corners. Since the pattern is only visible to cameras positioned in front of it, the calibration process requires more time to cover all camera views. To improve visibility from both directions, a glass chessboard [4] or a double-sided thick chessboard [5] has been proposed. Furthermore, during calibration, planar patterns are often tilted for simultaneous observation by multiple cameras, but tilting creates difficulties in detecting patterns, potentially reducing accuracy [6].

Unlike planar patterns, point-based methods using 3D markers such as spheres [7] provide a compelling alternative by ensuring consistent visibility across various camera viewpoints. Hence, for calibration in a large capture volume, professional motion capture systems, such as OptiTrack, Qualisys, and Vicon, prefer to use retro-reflective spherical markers tracked by infrared cameras. It is essential to highlight that the detection of these passive markers requires additional infrared or LED light sources positioned around the lens of each infrared or RGB camera. Although the use of spherical markers has been extended to RGB cameras through the use of colored spheres and color segmentation to extract the marker centroid [1], these colored spheres remain passive and require adequate ambient light for detection. Furthermore, they demand longer camera exposure, potentially leading to motion blur from the moving markers.

To address these challenges, we propose using an active calibration wand equipped with two LED-embedded markers similar to the work reported by Gossard et al. [6]. This method not only speeds up the extrinsic calibration process but also improves user-friendliness without compromising accuracy. The active wand can operate with a low camera exposure time, allowing it to move quickly during calibration. In addition, the known length of the wand allows for precise scale adjustments of the calibrated volume.

The wand is usually waved solely within the target capture volume. In some cameras, the projection of the wand marker will not spread across the entire field of view of the camera. Therefore, for the intrinsic calibration of each camera, multiple shots of a medium-sized checkerboard are used. These pre-calibrated intrinsic parameters can then be locked during the extrinsic bundle adjustment.

However, it has been observed that this standard intrinsic calibration approach often results in inconsistent focal lengths ($f_{x}$, $f_{y}$). This inconsistency indicates that locking all intrinsic parameters during the extrinsic bundle adjustment may not be ideal, as additional observations from the extrinsic calibration could improve the intrinsic parameters. On the other hand, allowing intrinsic parameters to change during extrinsic calibration can result in significant inaccuracies, especially when the wand’s field of view coverage is limited in some cameras. This presents a dilemma: while locking intrinsic parameters may lead to inaccuracies, allowing them to vary freely can result in even greater errors.

Therefore, we propose a new variant of the wand-based bundle adjustment method that incorporates checkerboard information from the intrinsic initialization step. This helps to constrain the intrinsic parameters across the camera’s field of view, including areas beyond the reach of wand waving.

The subsequent sections of this paper are organized as follows. Section 2 provides an overview of related work. Section 3 describes the details of our proposed calibration method and the design of tools. Section 4 presents the experimental approach, and results are discussed in Section 5. Section 6 discusses the limitations and future work. Finally, Section 7 concludes the paper.

## 2. Related Work

Calibrating multiple cameras is important for ensuring accurate performance in computer vision applications. This section reviews existing approaches, categorizing them based on their underlying principles and highlighting their strengths and limitations.

### 2.1. Calibration Using Planar Patterns

Camera calibration using planar patterns is widely adopted due to its ease of use and cost-effectiveness. The use of planar patterns with known geometries to facilitate keypoint detection is a well-established topic.

Early techniques involved a grid of black squares on a white background [3,8,9] and evolved to the use of black-and-white chessboard patterns supported by various toolboxes and the Open Source Computer Vision Library (OpenCV). To demonstrate the versatility of chessboard patterns, a single 5 by 6 chessboard with a square size of 120 mm, recognizable up to 6 m from cameras with VGA resolution (640 × 480 pixels), can be used for intrinsic, extrinsic, and floor calibration [10].

A combination of a standard chessboard pattern with ArUco fiducial markers [11] led to the development of the ChArUco board, which offers identifiable keypoints that enhance detection, even in cases of partial occlusion [2,12]. Alternatively, specially designed patterns with seemingly random features, coupled with descriptor-based keypoint detection, prove effective in calibrating both intrinsic and extrinsic parameters for multi-camera systems [13].

One limitation in using the planar pattern for extrinsic calibration lies in the restriction of movement speed during the camera’s exposure time, as rapid motion can result in blurry corners. Thus, to obtain images with clear patterns, the object is required to remain stationary, which leads to a longer calibration time, especially for a large capture volume with multiple camera perspectives.

### 2.2. Calibration Using Spherical Markers

Calibration using a spherical marker offers a rapid, versatile, and accurate approach for a multi-camera system, as it can be observed by cameras from all directions, unlike a planar pattern that can only be observed from a narrow range of angles in front of it. Furthermore, at least two valid observations of the calibration object from different perspectives are required for those frames to be used in the extrinsic calibration. Therefore, when calibrating with planar patterns, too many video frames are rejected when the pattern is viewed from behind or at a steep angle from a camera, making the calibration process less efficient.

Calibration with spherical markers can vary from a single marker to multiple markers. For instance, an LED light or a laser pointer with a small piece of colored plastic attached can be used to create a bright spot for visibility from different viewpoints [14,15]. However, methods that use only one marker are unable to determine the true scale of the waving volume, leading to uncertainty about the actual distance between cameras and the capture volume. This is because the cameras can be moved closer or farther away from the waving volume without changing observation or reprojection error if the waving volume is scaled accordingly.

To address the scaling issue, most methods opt for a wand with two or more markers at a known distance, such as two reflective spots on a rigid bar [16], a green and a red ball on a stick [1], or blinking LEDs inside three table tennis balls [6]. The 3D positions of the markers in each time frame are considered as a large set of unknown parameters that needs to be recovered, together with the camera parameters, using bundle adjustment optimization techniques.

A dual-marker wand commonly uses retro-reflective spherical markers, which requires each camera to have a ring of light sources around the lens. This configuration enables markers to reflect light directly back to the light sources (i.e., cameras) creating contrastively bright spots for efficient detection. During the optimization process, two common error terms to be minimized are 2D reprojection error from epipolar constraints [17] and wand-length error from Euclidean distance in 3D [17,18].

In addition to the dual-marker wand, which enhances volume coverage in calibration, it is common for calibration methods to incorporate a stationary calibration structure. This structure facilitates extrinsic initialization for numerical optimization and provides a way to define the origin, floor, and reference frame for subsequent measurements [17,18,19].

## 3. Calibration Method and Design of Tools

This section describes the step-by-step details of the proposed calibration method, as well as the comprehensive design of tools, formulas, and algorithms the calibration workflow. An overview of the entire calibration method is illustrated in Figure 1.

### 3.1. Ray Distance-Based (RDB) Triangulation

Ray distance-based (RDB) triangulation is an important element in both the calibration method and the subsequent evaluations. It differs fundamentally from the widely used direct linear transformation (DLT) method [20,21] in the derivation of its formula.

In the triangulation of each 2D observation point captured by a camera, there are two important rays. The *observation ray* is defined as a 3D ray that passes through the camera position and the back-projected 2D observation point. The *triangulated ray* is defined as a 3D ray that passes through the camera position and the triangulated 3D point. The DLT method is formulated by minimizing the magnitude of the cross product between the directions of the *observation ray* and the *triangulated ray* on each camera. Since these two rays always intersect at the camera position ($C_{i}$), minimizing the cross product essentially minimizes the angle between the two rays originating from the camera. The closer the camera is to the triangulated point (*P*) in 3D space, the greater the angle between the two rays. As a result, when a set of *observation rays* from a number of cameras is given, the 3D position result of DLT triangulation can be largely different depending on the combination of distances between the camera and the target object.

Given the sensitivity of DLT to the distance between the camera and the object of interest, this paper explores RDB triangulation as an alternative. The RDB triangulation formula is derived by minimizing the sum of squared distances between the *observation rays* from each camera and the triangulated point (*P*) in 3D [22]. Thus, RDB triangulation is invariant to the distances between cameras and the object of interest when the set of *observation rays* is the same.

#### 3.1.1. Ray Distance-Based (RDB) Formula

A triangulated 3D position (*P*) from RDB has a closed-form solution of
(1)$$\begin{array}{cc} P & ={\left(\sum_{i}Q_{i}\right)}^{-1}\left(\sum_{i}Q_{i}C_{i}\right)\\ \mathrm{and}\;Q_{i} & =I_{3}-U_{i}U_{i}^{⊤}, \end{array}$$
given that

$C_{i}$ is the 3D camera location associated with the $i^{th}$ *observation ray*;$I_{3}$ is the 3 by 3 identity matrix;$U_{i}$ is the 3D unit vector that represents the back-projected direction associated with the $i^{th}$ *observation ray*. This vector is calculated byUndistorting the 2D observation using OpenCV’s *undistortPointsIter* function with the normalized coordinate;Forming a 3D directional vector ${\left[x\_undistorted,y\_undistorted,1\right]}^{⊤}$ in the camera reference frame;Rotating the direction to the global reference frame using the current estimate of camera orientation;Normalizing the vector to obtain the unit vector ($U_{i}$).

#### 3.1.2. Definition of Ray Distance

Triangulating with *N observation rays* from *N* cameras, the term *ray distance* refers to the distance from the triangulated position in 3D to an *observation ray*. The square of this *ray distance* is formulated as follows:(2)$$\nu_{i}^{2}={\left(P-C_{i}\right)}^{⊤}Q_{i}\left(P-C_{i}\right)$$

### 3.2. Intrinsic Parameter Initialization

A rigid black-and-white chessboard with 10 by 7 internal corners (11 by 8 squares) and a square size of 35 mm is held still at 30 different poses in front of each camera similar to the images shown in Figure 2. The range of distances from a camera to the chessboard can vary depending on the angle of the field of view. As a reference, the distances from all cameras in our experiment are between 31 and 84 cm. The captured images are subsequently processed using standard computer vision techniques. OpenCV’s *findChessboardCorners* and *calibrateCamera* functions are used for corner detection and camera calibration, respectively. This workflow relies on Zhang’s method [3] to obtain the initial intrinsic calibration parameters. These parameters include 4 elements from the intrinsic matrix (i.e., $f_{x}$, $f_{y}$, $c_{x}$, $c_{y}$) and 5 parameters from the distortion coefficients (i.e., $k_{1}$, $k_{2}$, $p_{1}$, $p_{2}$, and $k_{3}$).

In the subsequent refinement steps detailed in Section 3.3.6, all the detected 2D chessboard corners, together with all the 6-DOF chessboard poses from this initial intrinsic calibration, are used again in our variant of bundle adjustment.

### 3.3. Extrinsic Calibration with a Dual-LED Wand

To perform extrinsic calibration and simultaneously refine the intrinsic parameters, the cameras must simultaneously observe a common object moving in the capture volume. In our approach, an active wand with dual LEDs is designed to serve this role.

#### 3.3.1. Wand Design

The idea of using two distinct colors—specifically, red and green tennis balls—for marker identification was introduced by Mitchelson and Hilton [1]. We improve the wand design by replacing passive colored markers with two markers that emit constant light of visible colors to enhance marker detection, identification, and localization. These light-emitting markers enable a very brief camera exposure time of 48 microseconds, effectively minimizing motion blur, even during fast wand movements. Additionally, their brightness improves the contrast between the markers and the background environment, making the wand more adaptable to a diverse range of environments.

Red and green colors are chosen for each marker to ensure distinctiveness while avoiding the large chromatic aberration observed in blue colors, as per our experimental findings. Each marker uses three LED filaments with a total power of 0.72 W enclosed within a diffusing, white, 3D-printed sphere measuring 20 mm in outer diameter. In contrast to markers of larger diameter, such as a 12-inch globe lamp [7], a 70 mm tennis ball [1], or a 40 mm table tennis ball [6], our markers are designed to be the smallest spheres that can fit the three LED filaments. The neck of each marker is designed to be as slim as possible, allowing just enough space for two electrical cables to pass through. This deliberate sizing reduces the chance of partial occlusion by the wand itself or the wand waver.

The two 3D-printed spheres are connected by a thin carbon-fiber rod. The measured distance between the centers of these spheres is 600 mm (to be used in Section 3.3.6 and Section 3.3.7). This length is large enough to keep the ratio of hand-made manufacturing error comfortably below 0.5% yet small enough to ensure portability with a detachable handle. With this length, both markers can be consistently observed simultaneously in the field of view of multiple cameras for a medium- to large-sized capture volume. The wand design is illustrated in Figure 3.

#### 3.3.2. Wand-Waving Procedure and Recording

The markers of the wand are intended to be swept through the entire target capture volume. To detect the floor plane within the capture space, a specific floor-touching component is also included in the waving procedure. While waving, the user is instructed to gently touch the red marker on six matte-black plates strategically positioned on the floor near various edges of the capture volume. These matte-black plates are designed to minimize reflections from the floor, mitigating potential inaccuracies in marker-center extraction. This floor-touching step facilitates floor detection without the need for additional calibration items. An example of the waving pattern is illustrated in Figure 4. The recommended waving duration can range from as short as one minute to longer durations, depending on the size of the capture volume.

All cameras record the waving motion at 10 frames per second with frame-level synchronization using shared trigger pulses. Each frame is recorded with an exposure time of 48 microseconds. At this frame rate, the subsequent marker localization step can be executed with a matched throughput, enabling calibration to be completed within 10 s after the completion of the wand-waving procedure.

#### 3.3.3. Static Noise Masking and 2D Marker Localization

This step aims to identify the 2D centers of visible red and green markers while excluding interference from other bright spots in the environment. Given that environments often contain bright elements such as light bulbs or computer screens visible in camera fields of view, pixels that are bright (intensity of red or green channel exceeding a threshold) for more than 5% of the recording period are masked out. This masking involves turning the intensity to zero in all channels for such pixels and their adjacent regions.

For each unmasked bright cluster of pixels in the image, the 2D center location (u,v) of all the bright pixels in the cluster is calculated in two ways to be conditionally selected.
Find the smallest circle that can enclose the bright pixel cluster and extract the center of this circle.Calculate the 2D centroid (the average position) of all the bright pixels within the cluster with an equivalent weight.
If the distance between the two candidates is greater than 0.5 pixels, there could be significant amounts of noise, such as partial occlusion, causing the 2D centroid from the second option to be unreliable. In this case, we choose the center of the circle from the first option as the answer. If the distance between the two candidates is not greater than 0.5 pixels, the 2D centroid from the second option is selected as the answer, as it usually provides better accuracy when the whole marker is seen.

For each video frame on each camera, up to one red and one green bright cluster with the largest sum of bright pixel intensity are used in the subsequent calculation.

#### 3.3.4. Dynamic Noise Rejection

The collected 2D center locations may contain noise from partially occluded markers, overlapping markers, or reflections of light on shiny surfaces. It is important to filter out such noise to ensure the accuracy and cleanliness of each marker’s trajectory.

First, frames adjacent to unidentified frames in each color sequence are discarded, as they are likely to be affected by partial occlusion. Next, approximated extrinsic matrices from all cameras are obtained using methods described in Section 3.3.5 and Section 3.3.7. Utilizing ray distance-based (RDB) triangulation (Section 3.1), these extrinsic matrices are used to triangulate and estimate the 3D locations of markers from all available 2D observations in every captured time frame. Any 2D observation associated with a *ray distance* exceeding 100 mm is deemed noise and rejected because it does not agree with the consensus from the rest of the cameras. This rejection is effective in eliminating random moving reflections, such as those from glasses or watch faces. Finally, by collectively examining all non-rejected markers, a linear relationship between ‘the number of bright pixels in the cluster’ ($\mu_{j}$) and ‘the inverse of the squared distance between the camera origin ($C_{i}$) and the triangulated point’ ($1/d_{j}^{2}$) is observed. This linear relationship can be described by Equation (Equation 3).
(3)$$\mu_{j}=\frac{\alpha_{i}}{d_{j}^{2}}+\beta_{i}$$
Therefore, the unknown parameters ($\alpha_{i}$ and $\beta_{i}$) for each camera can be determined through linear regression using all the data points obtained from the wand-waving step. Upon obtaining values for $\alpha_{i}$ and $\beta_{i}$, Equation (Equation 3) can be represented as the middle line in Figure 5. This line serves as the baseline and is shifted equally upwards and downwards to establish the boundary for noise rejection.

The upper boundary is chosen by sorting all the points with $\mu_{j}-\left(\frac{\alpha_{i}}{d_{j}^{2}}+\beta_{i}\right)$ in descending order and selecting the sixth point. Given that the sixth point has $\eta_{i}=\mu_{j}-\left(\frac{\alpha_{i}}{d_{j}^{2}}+\beta_{i}\right)$, this $\eta_{i}$ is used to reject any points when $m_{j}>\frac{\alpha_{i}}{d_{j}^{2}}+\beta_{i}+\eta_{i}$ (exceeding the upper bound) or $\mu_{j}<\frac{\alpha }{d_{j}^{2}}+\beta_{i}-\eta_{i}$ (falling below the lower bound). The reason for choosing the sixth point is that outliers typically do not occur on the positive side (the marker size cannot become larger by itself). On the other hand, marker observations can become smaller due to partial occlusion by the wand itself or the body parts of the wand waver.

After the rejection process, the result consists of the cleaned 2D locations (〈u,v〉) of all surviving observations, which are labeled as either green or red.

#### 3.3.5. Extrinsic Initialization

Extrinsic initialization involves calculating the relative transformation (6 DOF) between pairs of cameras and using this information to link all the cameras within a single reference frame. First, a random camera is assigned as the base camera with an identity homogeneous transformation matrix as its extrinsic matrix (camera pose). The initial position and orientation of this base camera are used as a global reference frame for all the calculations throughout the calibration process until a user-defined reference frame is created in Section 3.3.9. Subsequently, the process involves iterating through the remaining cameras and acquiring the transformation matrix with respect to the base camera. For each pair of cameras, this is achieved by gathering a list of all 2D–2D correspondence pairs where both marker observations have not been previously rejected. Following this, OpenCV’s *undistortPointsIter* function is used to undistort the 2D points for each camera using their respective intrinsic matrix and distortion values. Then, OpenCV’s *findEssentialMat* function is used with the undistorted correspondence pairs to estimate the possible essential matrices. Since the *findEssentialMat* function could return more than one essential matrix, the selection process involves choosing the essential matrix that best fits all available points. Given that $y^{\prime }$ and *y* represent an undistorted 2D correspondence pair (observation from the same 3D point) in homogeneous and normalized form, the essential matrix (*E*) is defined as
(4)$$y^{\prime ⊤}Ey=0$$
Therefore, the most suitable essential matrix is identified based on the matrix that minimizes $\sum_{i}\left|y_{i}^{\prime ⊤}Ey_{i}\right|$. Finally, OpenCV’s *recoverPose* function is used to extract the extrinsic rotation matrix from the selected essential matrix.

#### 3.3.6. Optimization with Bundle Adjustment

The initial values obtained through the implementation of Section 3.2 and Section 3.3.5 may not be precise. Therefore, this section aims to refine the parameters of both the extrinsic and intrinsic matrices for all cameras using our variant of the bundle adjustment optimization technique.

The Levenberg–Marquardt (LM) optimization algorithm is used to iteratively update a set of unknown parameters ($\vec{\theta }$) so that all the values in a residue vector ($\vec{e}=F\left(\vec{\theta }\right)$) become as close as possible to zero.

The unknown parameters ($\vec{\theta }$) consist of the following values in order.

Each camera (with index *i*) has 6 extrinsic parameters representing its pose with the translation and orientation of the global reference frame relative to the camera reference frame (orientation in axis-angle representation: $\vec{r_{i}}=\langle r_{xi},r_{yi},r_{zi}\rangle$; translation: $\vec{t_{i}}=\langle t_{xi},t_{yi},t_{zi}\rangle$) and 9 intrinsic parameters (focal length: $f_{xi}$, $f_{yi}$; optical center: $c_{xi}$, $c_{yi}$; distortion coefficients: $k_{1i}$, $k_{2i}$, $p_{1i}$, $p_{2i}$, and $k_{3i}$). To simplify subsequent explanations, all 9 intrinsic parameters for camera *i* are written as $\vec{s_{i}}$.Each frame (with index *j*) contains the 3D positions of the green marker ($\vec{o_{1j}}=\langle o_{1xj},o_{1yj},o_{1zi}\rangle$) and red marker ($\vec{o_{2j}}=\langle o_{2xj},o_{2yj},o_{2zj}\rangle$).Each camera previously took 30 or fewer images of the chessboard in different poses to initialize intrinsic calibration. Each image of the chessboard (with board index *b*) has 6 unknown chessboard pose parameters in its camera reference frame (board orientation in axis-angle representation: $\vec{\zeta_{b}}=\langle \zeta_{xb},\zeta_{yb},\zeta_{zb}\rangle$; board translation: $\vec{\phi_{b}}=\langle \phi_{xb},\phi_{yb},\phi_{zb}\rangle$).

All these unknown parameters can be initialized by steps previously explained in Section 3.2 and Section 3.3.5. RDB triangulation (Section 3.1) is used to initialize 3D marker positions.

The residue vector ($\vec{e}=F\left(\vec{\theta }\right)$) consists of the following values in order.

For a camera with index *i* at frame index *j*, 4 residual values are calculated. Two values are from the signed difference between the 2D projection from the green marker’s 3D position ($\vec{o_{1j}}$) and the observed 2D location ($\langle u_{1ij},v_{1ij}\rangle$). The other two values are from the red marker. This can be formulated as
(5)$$\langle △u_{1ij},△v_{1ij}\rangle =\langle u_{1ij},v_{1ij}\rangle -Projection\left(\vec{r_{i}},\vec{t_{i}},\vec{s_{i}},\vec{o_{1j}}\right).$$The Projection function follows the standard OpenCV formulation of projecting 3D points to an image plane (using OpenCV’s *projectPoints* function). Note that for any observation that is flagged as noise according to Section 3.3.4 or when no observation available in a frame, the residue values associated with that projection are always zero throughout optimization.For each chessboard index (*b*), each corner (*w*) on the chessboard is projected to the image plane of the corresponding camera (*i*) for comparison with the observed corner in 2D ($\langle u_{bw},v_{bw}\rangle$) on camera *i*. This can be formulated as
(6)$$\langle △u_{bw},△v_{bw}\rangle =\langle u_{bw},v_{bw}\rangle -Projection\left(\vec{\zeta_{b}},\vec{\phi_{b}},\vec{s_{i}},\vec{\tau_{w}}\right),$$
given that $\tau_{w}$ is the static 3D position of corner *w* of the chessboard in the chessboard reference frame.

If a calibration session involves *N* cameras, *M* frames, *B* total chessboard images, and *W* corners on a chessboard, the length of the input vector ($\vec{\theta }$) is 15N+6M+6B, while the residue vector ($\vec{e}$) has a length of 4NM+2BW. The size of the Jacobian matrix (*J*) of function *F* used in the LM algorithm is (4NM+2BW) by (15N+6M+6B). An example of the Jacobian matrix structure with a set of small hyperparameters (N=3, M=8, B=9, and W=6) is illustrated in Figure 6.

To calculate this large Jacobian matrix accurately in a practical amount of time, PyTorch’s forward-mode automatic differentiation is used as the main tool. Equations (Equation 5) and (Equation 6) are implemented as a computational graph that wraps OpenCV’s *projectPoints* function in PyTorch to make the entire computational graph automatically differentiable.

In each round of forward-mode automatic differentiation, the tangent of multiple input parameters can be set to 1 simultaneously, as those input parameters do not have overlapping influence on any of the residue values.
Round 1: The tangents of all $r_{xi}$ and all $\zeta_{xb}$ (total of N+B parameters) are set to 1.Round 2: The tangents of all $r_{yi}$ and all $\zeta_{yb}$ (total of N+B parameters) are set to 1.Round 3: The tangents of all $r_{zi}$ and all $\zeta_{zb}$ (total of N+B parameters) are set to 1.Round 4: The tangents of all $t_{xi}$ and all $\phi_{xb}$ (total of N+B parameters) are set to 1.Round 5: The tangents of all $t_{yi}$ and all $\phi_{yb}$ (total of N+B parameters) are set to 1.Round 6: The tangents of all $t_{zi}$ and all $\phi_{zb}$ (total of N+B parameters) are set to 1.Round 7: The tangents of all $f_{xi}$ (total of *N* parameters) are set to 1.Round 8: The tangents of all $f_{yi}$ (total of *N* parameters) are set to 1.Round 9: The tangents of all $c_{xi}$ (total of *N* parameters) are set to 1.Round 10: The tangents of all $c_{yi}$ (total of *N* parameters) are set to 1.Round 11: The tangents of all $k_{1i}$ (total of *N* parameters) are set to 1.Round 12: The tangents of all $k_{2i}$ (total of *N* parameters) are set to 1.Round 13: The tangents of all $p_{1i}$ (total of *N* parameters) are set to 1.Round 14: The tangents of all $p_{2i}$ (total of *N* parameters) are set to 1.Round 15: The tangents of all $k_{3i}$ (total of *N* parameters) are set to 1.Round 16: The tangents of all $o_{1xj}$ and all $o_{2xj}$ (total of 2M parameters) are set to 1.Round 17: The tangents of all $o_{1yj}$ and all $o_{2yj}$ (total of 2M parameters) are set to 1.Round 18: The tangents of all $o_{1zj}$ and all $o_{2zj}$ (total of 2M parameters) are set to 1.
With this trick, the calculation can be reduced from 4NM+2BW rounds of reverse-mode automatic differentiation to just 18 rounds of forward-mode automatic differentiation. In our experiment, this trick can improve the Jacobian calculation speed by over 1000 times compare to the use of reverse-mode automatic differentiation to fill the whole matrix (*J*).

The LM algorithm calculates the parameter update in each iteration by solving the following linear system to find the update ($\vec{\delta }$).
(7)$$\left(J^{⊤}J+\lambda I\right)\vec{\delta }=J^{⊤}F\left(\vec{\theta }\right)$$

Equation (Equation 7) is a large linear system with $J^{⊤}J+\lambda I$ as a square matrix and a dimension of 15N+6M+6B that can take time to directly solve. However, since the majority of matrix $J^{⊤}J+\lambda I$ is zeros with a deterministic structure, as illustrated in Figure 7, the Schur complement trick can be used to solve for $\vec{\delta }$ more efficiently. To simplify the explanation, Equation (Equation 7) is arranged into the following form.
(8)$$\left[\begin{array}{cc} A & B\\ B^{⊤} & D \end{array}\right]\left[\begin{array}{c} \delta_{1}\\ \delta_{2} \end{array}\right]=\left[\begin{array}{c} q_{1}\\ q_{2} \end{array}\right]$$

Matrix A, with size 15N×15N, is a block-diagonal matrix that contains *N* blocks of size 15 × 15.Matrix D, with size (6M+6B)×(6M+6B), is also a block-diagonal matrix that contains 2M blocks of size 3 × 3 and *B* blocks of size 6 × 6.Matrix B, with size 15N×(6M+6B), is treated as a dense matrix.Vectors $\delta_{1}$ and $q_{1}$ have the same length of 15N.Vectors $\delta_{2}$ and $q_{2}$ have the same length of 6M+6B.

With the Schur complement trick, $\delta_{1}$ can be solved using a much smaller linear system.
(9)$$\left(A-BD^{-1}B^{⊤}\right)\delta_{1}=q_{1}-BD^{-1}q_{2}$$
Because the inverse of D can be calculated rapidly by independent inversion of each small block with dimensions of 3 × 3 and 6 × 6 along the diagonal line of D, this trick can offload a lot of computation.

After $\delta_{1}$ is known, $\delta_{2}$ can be calculated by
(10)$$\delta_{2}=D^{-1}\left(q_{2}-B^{⊤}\delta_{1}\right).$$

Then, $\delta_{1}$ and $\delta_{2}$ are concatenated as an update vector ($\vec{\delta }$). If the update causes the average squared residue to reduce (e.g., $\sum F{\left(\vec{\theta }+\vec{\delta }\right)}^{2}<\sum F{\left(\vec{\theta }\right)}^{2}$), the unknown state $\vec{\theta }$ is updated to $\vec{\theta }+\vec{\delta }$, and λ is reduced by half in the next iteration. Otherwise, if $\vec{\delta }$ does not improve the average squared residue, λ is doubled, and Equation (Equation 7) is solved again until $\vec{\delta }$ causes the average squared residue to reduce. The λ parameter is initiated from 0.25, and the iteration is stopped after λ is increased to $2^{24}$.

#### 3.3.7. Scale Correction Using Wand Length

Without scale correction, the average 3D distance between the triangulated red and green markers (Average_wand_length) from all the frames is different from the intended wand length (Intended_wand_length, e.g., 600 mm in our experiment). To correct the scale of the calibrated space to match reality, the translation vector associated with each camera ($\vec{t_{i}}$) must be scaled by a ratio of Intended_wand_length over Average_wand_length. Re-triangulation of the active wand markers with the modified $\vec{t_{i}}$ makes all the triangulated position vectors scale by the same ratio, causing the average wand length to match the intended wand length.

#### 3.3.8. Ground-Plane Detection by Floor Touching

Ground-plane detection is an essential part of many 3D applications, such as motion capture for biomechanical analysis or animation. Typically, floor calibration involves having pre-calibrated cameras observing a calibration object such as a chessboard [10] or a 3D structure of markers [17] placed on the floor. However, our method can recover the ground plane without the need for an additional calibration object. During wand waving in the capture volume, the user simply needs to gently touch a wand marker on at least four different spots on the floor of the capture volume—specifically, on predetermined matte-black plate locations.

A 3D convex hull is generated from the set of all triangulated markers. The triangle face on the convex hull with the largest number of triangulated points that fall within 10 mm from its expanded plane are selected as the *stem face*. Any triangulated point that is a part of the convex hull and falls within 10 mm of the expanded plane of the *stem face* is collected as a floor-touching point. For any space of a 100 mm-diameter sphere, only one floor-touching point that is closest to the expanded plane of the *stem face* is allowed.

Eventually, only four to six floor-touching points are detected. The third principal component of these samples is used as the normal vector of the floor (i.e., the Z axis of the capture volume), and the average of these samples offset by the active marker radius and the black plate’s thickness is used to define the floor level.

In contrast to using a relatively small floor calibration object that can easily produce a few degrees of tilt error, especially on an uneven floor, the floor-touching method is designed to minimize the unwanted tilt of the detected floor plane. Strategically placing the floor-touching spots near the edges of the capture area is similar to having a large calibration object that covers the entire floor of the capture area, therefore producing less tilt error.

#### 3.3.9. Optional L Frame to Define the X and Y Axes

In cases where the user needs to define the origin or the direction of the X and Y axes of the capture volume, an additional record is required using an L frame with 4 active red markers (Figure 8).

From each camera perspective, the marker identities (L1, L2, L3, and L4) are assigned by detecting the three collinear markers (L1, L2, and L3) and determining their distances. This identification for allows the triangulation of all four markers, which can be used to calculate the origin (L0 projected to the floor plane) and the direction of the X axis (L1–L3 projected to the floor plane) and Y axis (with a cross product of the Z and X axes). The normal vector and the level of the ground plane are not changed by these calculations because the new origin is still on the ground plane, and the directions of the X and Y axes are always parallel to the ground plane.

Without the use of an L frame, the system will project all the triangulated wand markers to the ground plane, identify the direction with the largest variance (the first principal component) using principal component analysis, and assign that direction as the X axis.

The transformation of the coordinate reference frame from the frame described in Section 3.3.7 to a new reference frame with a new origin and orientation is applied to all the camera poses (extrinsic matrix) as the last step.

## 4. Evaluation Experiment

To benchmark the proposed calibration method’s accuracy, it is compared to a method assisted by a marker-based motion capture system (Section 4.3), a chessboard-based method used in Anipose [2] (Section 4.4), and several variants of our method (Section 4.5).

### 4.1. Space and Camera Setup

As illustrated in Figure 9, the target capture volume with a height of 2 m is surrounded by 7 RGB cameras positioned approximately 1.7 m above the ground. Each camera (See3CAM_24CUG, e-con Systems^®^, Chennai, India) is a global shutter RGB camera with a resolution of 1920 × 1200 pixels. All cameras are in landscape orientation. Fitted with a varifocal lens, each camera is adjusted to encompass a field of view just large enough to see the entire capture volume from its perspective. Positioned around the tip of the lens are three 1-W LEDs (Figure 10) designed to emit constant white light whenever a passive retro-reflective marker is used in the experiment. The horizontal distance from a camera to the center of the capture volume ranges from around 3.7 m to 6.6 m. Throughout the entire experiment, none of the cameras are touched or moved.

### 4.2. Marker-Based Motion Capture (Mocap) System

In addition to the RGB cameras, a marker-based mocap system is also used in the experiment. This system comprises 10 high-resolution infrared cameras (Qualisys Arqus A12, 12 MP) and 6 medium-resolution infrared cameras (Qualisys Miqus M3, 2 MP). The main functions are to assist one of the calibration methods and generate the final 3D trajectory as a ground truth to benchmark the accuracy of each calibration method. The mocap system is calibrated once before the start of the entire evaluation experiment.

### 4.3. Calibration Method Assisted by Marker-Based Mocap

This method uses external information from the marker-based mocap system to assist in finding the set of calibration parameters for the 7 RGB cameras. Although this method is not considered a practical stand-alone calibration method, it serves as an effective means to establish an upper-bound performance in the comparison of calibration methods, as this is the only method that has access to external 3D information.

A retro-reflective marker with a diameter of 19 mm positioned at the tip of a wand is waved inside the capture volume. The marker’s movements are simultaneously recorded by both the marker-based mocap system and all video cameras that receive synchronization pulses from the mocap system for frame-based synchronization.

Given that the mocap system records the 3D trajectory of the marker at a rate of 200 Hz, the data are downsampled to match the video sampling rate of 50 Hz. This downsampling is applied through linear interpolation at the midpoint of each exposure period of the RGB cameras. All RGB cameras use the same exposure time of 996 microseconds. The marker, appearing as a bright spot on the RGB image, generates a series of 2D center locations (*u*,*v*) extracted at 50 Hz in each RGB camera.

For each RGB camera, all valid 2D–3D correspondence pairs are collected and fed into OpenCV’s *calibrateCamera* function to find all the intrinsic and extrinsic camera parameters. Note that the intrinsic parameters are initialized using the method described in Section 3.2. This mocap-assisted calibration method is used to ensure a good spatial alignment between video cameras in the marker-based motion capture reference frame for subsequent comparison with ground truth from marker-based mocap in our previous study [23].

### 4.4. Anipose’s Calibration Method

Anipose is a toolkit for robust markerless 3D pose estimation [2] that also includes a calibration tool for multi-camera systems. To use Anipose’s calibration method, a ChArUco board with 4 by 5 squares and a square size of 150 mm is printed and attached to a rigid metal frame (Figure 11c). The large square size allows large ArUco markers with a length of 112.5 mm to fit inside, ensuring visibility by the cameras, even from a considerable distance.

All RGB cameras are synchronized at the frame level using a common pulse generator, recording videos at 50 Hz. The ChArUco board is slowly moved within the capture volume to be seen by all cameras. To ensure adequate light has been captured by the cameras, the exposure period is set to 3.89 ms, while the measured illuminance in the area is in the range of 520 to 580 lux. After the recording, the calibration process is fully automated, requiring no human intervention.

Behind the scenes, Anipose’s calibration uses a bundle adjustment technique to simultaneously recover camera parameters and board poses in all time frames through an optimization process.

### 4.5. Variants of Wand-Based Bundle Adjustment

Based on our optimization method presented in Section 3.3.6, which uses the active wand as the main calibration object, various versions of bundle adjustment are also tested to provide better understanding in the analysis of the proposed method.

**INIT**: The common initialization state shared across all bundle adjustment variants. This state is benchmarked to observe the improvement from each optimization.**FusedBA**: Uses the optimization method proposed in Section 3.3.6 with a chessboard constraint. Each camera has 15 adjustable parameters (6 extrinsic and 9 intrinsic parameters).**BAp15**: Removes the chessboard-related parameters and residues from the optimization. Each camera has 15 adjustable parameters (6 extrinsic and 9 intrinsic parameters).**BAp10**: Same as BAp15, but each camera has 10 adjustable parameters. All 5 lens distortion parameters are locked to the initial state during bundle adjustment.**BAp6**: Same as BAp15, but each camera has 6 adjustable extrinsic parameters. All 9 intrinsic parameters are locked to the initial state during bundle adjustment.**FusedBA+W** and **BAp15+W**: Similar to FusedBA and BAp15, respectively, but a wand-length constraint is added to the optimization by inserting the difference between the intended wand length and the modeled wand length as a residue in every frame. This residue can be formulated as
(11)$$△l_{j}=Intended\_wand\_length-\left|\right|\vec{o_{1j}}-\vec{o_{2j}}\left|\right|$$
for a frame with index *j* in mm. Owing to the insertion of these *M* additional residues into the residue vector ($\vec{e}$), the structure of the Jacobian matrix differs from that shown in Figure 6, and the forward-mode automatic differentiation in rounds 16, 17, and 18 must replaced by 6 rounds (one round each for $o_{1xj}$, $o_{2xj}$, $o_{1yj}$, $o_{2yj}$, $o_{1zj}$, and $o_{2zj}$). Also, in the subsequent calculation, the upper-left structure of matrix D changes from 2M diagonal blocks of size 3 × 3 to *M* diagonal blocks of size 6 × 6. These two variants are included in the experiment to test the impact of the wand-length constraint.

### 4.6. Benchmarking Workflow

Each of the aforementioned calibration methods is repeated 33 times, producing 33 sets of calibration parameters for all 7 cameras (33 for the mocap-assisted method, 33 for Anipose, and 33 for all 7 variants of active wand-based methods). To evaluate the accuracy of each set of calibration parameters, a single benchmarking record is created for common use across all 231 sets of calibration parameters.

During the benchmarking record, a retro-reflective marker is waved for 2 min inside the capture volume using an extended rod from outside the capture volume to minimize marker occlusion. This waving sequence is recorded with frame-level synchronization between the marker-based mocap system and the 7 RGB cameras. The marker-based system records the 3D trajectory of the marker at 200 Hz, while the 7 RGB cameras record images at 50 Hz with an exposure time of 996 microseconds. As the marker appears as a bright spot in the RGB image, a series of 2D center locations (*u*,*v*) is extracted at 50 Hz. The time offset between the marker-based mocap frame and the midpoint of the RGB camera exposure period is measured and used in the interpolation of the marker-based data to downsample the data from 200 Hz to 50 Hz.

With any set of calibration parameters, all the 3D positions of the marker can now be triangulated from the pre-calculated 2D center locations for comparison against corresponding 3D locations from the marker-based system. Since not all the calibration methods’ outputs are in the same reference frame as the marker-based system, each triangulated marker trajectory is rigidly transformed [24] to align with the marker-based system’s trajectory before calculating the Euclidean distance between each corresponding pair of 3D points to determine the average error.

This way of benchmarking against a ground-truth 3D trajectory from an independent record has not been found in similar calibration studies. Typically, calibration studies rely on the reprojection errors from the same record used for calibration [1,6,15,25,26,27]. However, this conventional benchmarking method presents three issues. First, there is a risk of overfitting to the calibration record, and the result may not generalize well to subsequent records that use those calibration parameters. Secondly, the reprojection error is often scale-invariant and does not adequately reflect errors related to the scale of the calibrated coordinate frame. Lastly, the magnitude of the reprojection error depends largely on the focal length (zoom level) of each camera and the distance from the capture volume. These dependencies make the report of reprojection error not very meaningful, especially when all the cameras are configured very differently. Muglikar et al. [28] also highlighted the inadequacy of relying solely on reprojection error in assessing the quality of a calibration.

Therefore, we propose the use of the error of triangulated 3D positions against a gold standard from an independent record as the main metric. This approach provides a more insightful evaluation, reflecting the actual use of the system for 3D application without the risk of overfitting. The whole benchmarking workflow is summarized in Figure 12.

### 4.7. RDB vs. DLT Triangulation

With a hypothesis that RDB triangulation is more accurate than DLT triangulation, in this study, the triangulation step in the benchmarking method is performed twice with the two different triangulation methods. This results in two different lists of errors for each calibration method, serving to validate the hypothesis.

### 4.8. Additional Restriction

For each calibration round, the record length is limited to 1 min to ensure a fair and consistent comparison across different calibration methods.

### 4.9. Realistic Calibration Dataset

All the records of calibration and the benchmark in this study represent a unique dataset that can be used to develop a realistic multi-camera calibration algorithm. Unlike simulated data, this dataset contains realistic forms of noises, such as partial occlusion from the wand or the waver; chromatic aberration; lens imperfection; color inaccuracy from the demosaicing process resulting from the Bayer filter pattern; artifacts from video compression; or focal blur during the snapshot of the near-range chessboard as the lenses are focusing on the capture volume, which is far away. All the files are made publicly available as the *MCalib* dataset at koonyook.github.io/MCalib (accessed on 15 November 2024) for research purposes (Use passcode b875UZ to download) so that new methods can be compared and improved further under a standardized benchmark.

## 5. Results and Discussion

### 5.1. Accuracy and Consistency

As shown in Table 1, the mocap-assisted calibration method achieves the highest accuracy based on average errors from our benchmarking method. It is closely followed by our proposed method (FusedBA), while Anipose’s calibration method has the lowest level of accuracy.

Among all the stand-alone calibrations, our method (FusedBA) versus Anipose’s calibration shows 78.7% and 79.1% reductions in the average 3D errors with RDB and DLT triangulation, respectively. These results are achieved even though our FusedBA method operates at one-fifth of the frame rate used in Anipose’s calibration. Furthermore, the standard-deviation values highlight different orders of magnitude in consistency: at the micrometer level for mocap-assisted calibration, the sub-millimeter level for FusedBA, and a few millimeters for Anipose’s calibration.

These results agree with the examples of qualitative results presented in Figure 13, where the frustum projection from an opposite camera tends to have worse alignment with the actual image when the calibration accuracy is lower.

All the variants of bundle adjustment methods show significant improvements in both accuracy and consistency relative to the initialization state (INIT). A more detailed distribution of the mean errors from all 33 rounds of calibration is plotted in Figure 14. The performances of BAp6 and BAp10 are virtually the same, with large improvements in consistency. For RDB triangulation, their mean errors and standard deviations drop from the initialization state (INIT) by 4.2% and 97.1%, respectively. The fact that BAp10 does not achieve any improvement relative to BAp6 could imply that partial intrinsic adjustment of the focal length and optical center without allowing for the simultaneous adjustment in lens distortion parameters does not cause any improvement in the intrinsic parameters because they are all entangled and need to be improved together. This phenomenon is more obvious with the observable improvements in both the BAp15 and FusedBA methods that allow all the intrinsic parameters to be optimized. The BAp15 and FusedBA methods can reduce the error even further, with drops of 5.4% and 12.9% relative to INIT, respectively.

One interesting observation among the bundle adjustment variants is in the rightmost column of Table 1. This column shows the means and standard deviations of RMS projection errors of the wand markers used in the calibration record after the optimization terminates. This metric is commonly used to report the accuracy of calibration in the literature [1,6,15,25,26,27]. Although all the variants improved largely from the INIT state, being the lowest in this metric does not translate to the best 3D accuracy on the benchmark record. Specifically, for BAp15, when the lens distortion coefficients are allowed to update when the wand coverage cannot practically cover the entire camera field of view, optimization can overfit easily in the wand-waving area in the image and ignore any negative side effect of incorrect distortion in other parts of the image. An example of this phenomenon is shown in Figure 15. Even though this overfitting effect is not seen in the BAp6 or BAp10 variants, downgrading from BAp15 to BAp6 or BAp10 is a loss of opportunity for the use of additional observations from the wand-waving step to refine all the intrinsic parameters. This rationale underpins our proposal of the FusedBA method, which enables fine tuning of all intrinsic parameters while preventing the overfitting issue.

Comparing the results from BAp15+W against those of BAp15 and those from FusedBA+W against those of FusedBA shows that adding a wand-length constraint to the optimization does not significantly improve the benchmarking results in both comparisons. Instead, it slightly increases the average errors in all the comparisons. This could be caused by a few possible issues. The first issue is the inability to accurately measure the distance between the two wand markers to a sub-millimeter level. The second possible issue is the inability to produce wand markers that evenly diffuse light on the marker surface without any shadow so that the marker appears perfectly round from all perspectives. These reasons may prevent the wand-length constraint from working properly. Nonetheless, the scale correction step via camera translation adjustment (Section 3.3.7) can provide well-adjusted triangulation results without relying on the wand-length constraint.

### 5.2. Accuracy of Scale

In the alignment calculation step described in Section 4.6, in addition to determining translation and rotation terms, a scaling factor is also computed, as presented in Table 2. This scaling factor serves as the multiplier for the entire capture volume to optimally minimize differences between the triangulated 3D trajectory and the ground-truth 3D trajectory from the marker-based benchmark record. A scaling factor of 1.0 means that the size of the triangulated 3D trajectory perfectly matches the ground-truth trajectory. According to the scaling factors, with an RDB triangulated trajectory, the average scales of calibrated space from the Anipose, mocap-assisted, INIT, BAp6, BAp10, BAp15, BAp15+W, FusedBA, and FusedBA+W methods are different from the ground truth by 0.4807%, 0.0011%, 0.0429%, 0.0007%, 0.0007%, 0.0196%, 0.0214%, 0.0033%, and 0.0056%, respectively.

### 5.3. Is RDB More Accurate than DLT Triangulation?

In every calibration method, RDB triangulation consistently yields significantly fewer 3D errors compared to the DLT method. However, the level of significance is relatively lower in Anipose’s calibration method, where the calibration parameters contain more noise. Additional details are shown in Table 3.

### 5.4. Computation Time

As most of the operations involve sparse matrices with known deterministic structures of zero elements, a careful implementation using the NumPy library can be used to avoid zeros and reduce the calculation time of the FusedBA method. Among the 33 rounds of calibration, the total LM optimization takes 3.83 s, on average (ranges from 2.93 to 5.14 s), on an Intel^®^ Core^TM^ i9-13900HK processor. The parameters are updated 7.03 times, on average (range from 5 to 10 times), before the optimization converges. Each round of LM optimization from the start until successful parameter update takes 0.54 s to process, on average. The part that takes the maximum amount of time in this method is the calculation of the Jacobian matrix (*J*), which takes 80.42% of the entire processing time.

## 6. Limitation and Future Work

Without specialized tools such as a a laser interferometer or a large micrometer, we could not obtain an accurate measurement of the distance between the centers of the two markers on the wand. It is also difficult to arrange the light sources within each marker to diffuse evenly on the surface of the marker without any shadow. This might cause the roundness of the marker to be perceived slightly differently from different perspectives. Therefore, the wand-length constraint is not recommended to be included in our variant of bundle adjustment if the true length of the wand is not precisely measured. A better wand design with precise manufacturing techniques may improve the method in these areas.

As the accuracy gap between the FusedBA method and the mocap-assisted calibration is still significant, the proposed method could be improved further by exploring better extrinsic initialization through the use of different orders linking camera pairs into the same reference frame.

In terms of execution speed, as the Jacobian matrix calculation is the biggest bottleneck in the processing time and it contains repeating calculations of forward-mode automatic differentiation, vectorization of the calculation graph in a graphical processing unit (GPU) could accelerate the algorithm further. If GPU acceleration is much faster, it may allow this algorithm to explore optimization from various initialization states to reach better local minima.

## 7. Conclusions

We propose a new variant of the wand-based bundle adjustment method that also utilizes the chessboard information from the intrinsic calibration step to help constrain the intrinsic parameters throughout the camera field of view, inclusive of the area where wand waving cannot reach. This method practically and effectively prevents overfitting in the optimization process, providing consistent and more accurate results on the 3D benchmark.

## Figures and Tables

**Figure 1 sensors-24-07416-f001:**
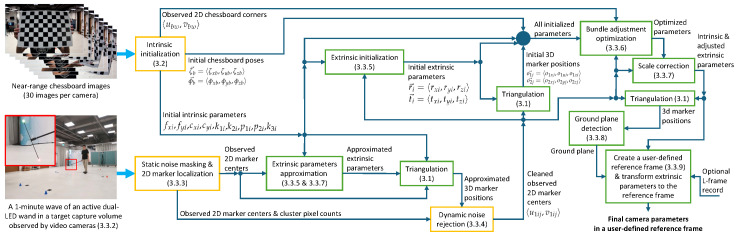
A schematic diagram of the proposed calibration method. Each orange box represents an operation that is performed independently on each camera. Each green box represents an operation that is performed together across all the cameras. The arrows represent the way that information flows between operations.

**Figure 2 sensors-24-07416-f002:**
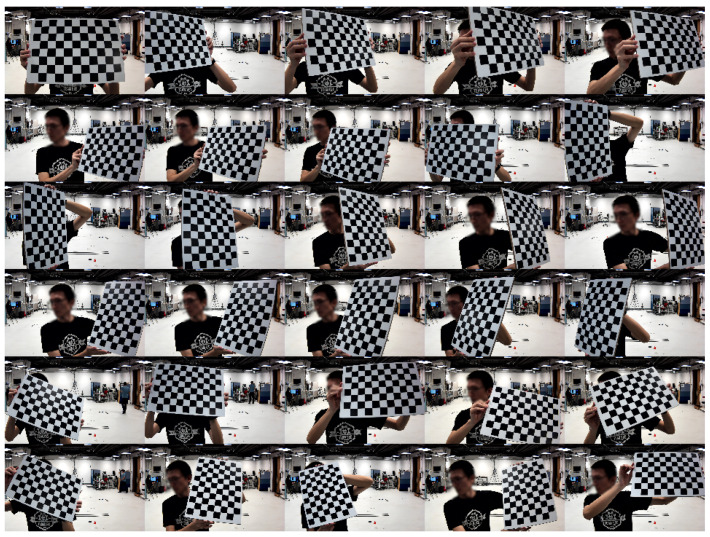
An example of a set of 30 chessboard images from a camera used in both intrinsic initialization and subsequent bundle adjustment (FusedBA only). In this step, the collection of chessboard corners can easily spread across the entire image area from a near range, as this step is performed exclusively on each camera. The user does not need to try making the chessboard visible to be registered by more than one camera.

**Figure 3 sensors-24-07416-f003:**
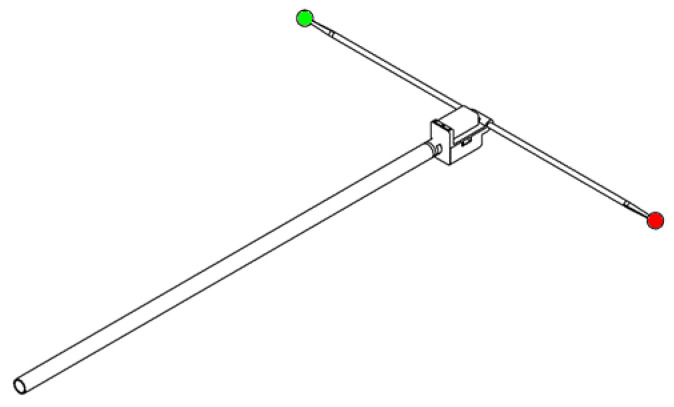
Calibration wand with red and green active markers.

**Figure 4 sensors-24-07416-f004:**
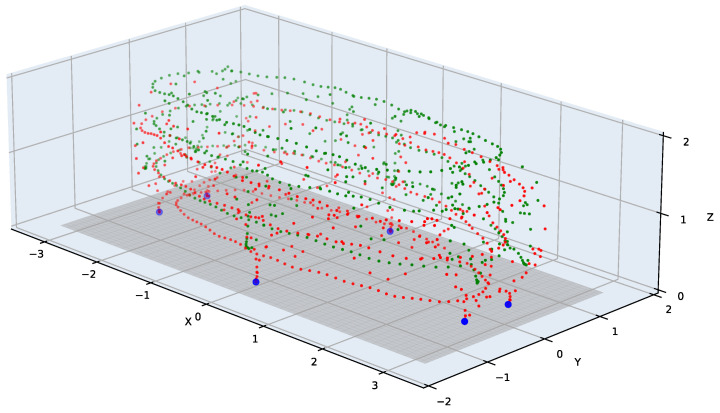
An example of the waving pattern using the active wand. The red and green points are the red and green active marker positions from all the time frames, respectively. The six blue points are the detected floor-touching points that are used to fit the floor plane (gray rectangle).

**Figure 5 sensors-24-07416-f005:**
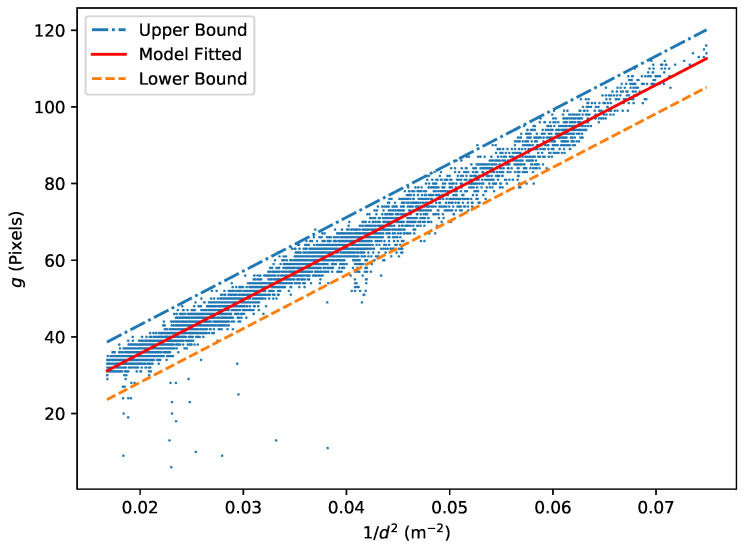
An example plot of the number of bright pixels in a cluster (*g*) against the inverse of the squared distance from the triangulated position to the camera ($1/d^{2}$). Without the noises under the lower boundary, the distribution is fairly symmetrical on both sides of the fitted linear regression model.

**Figure 6 sensors-24-07416-f006:**
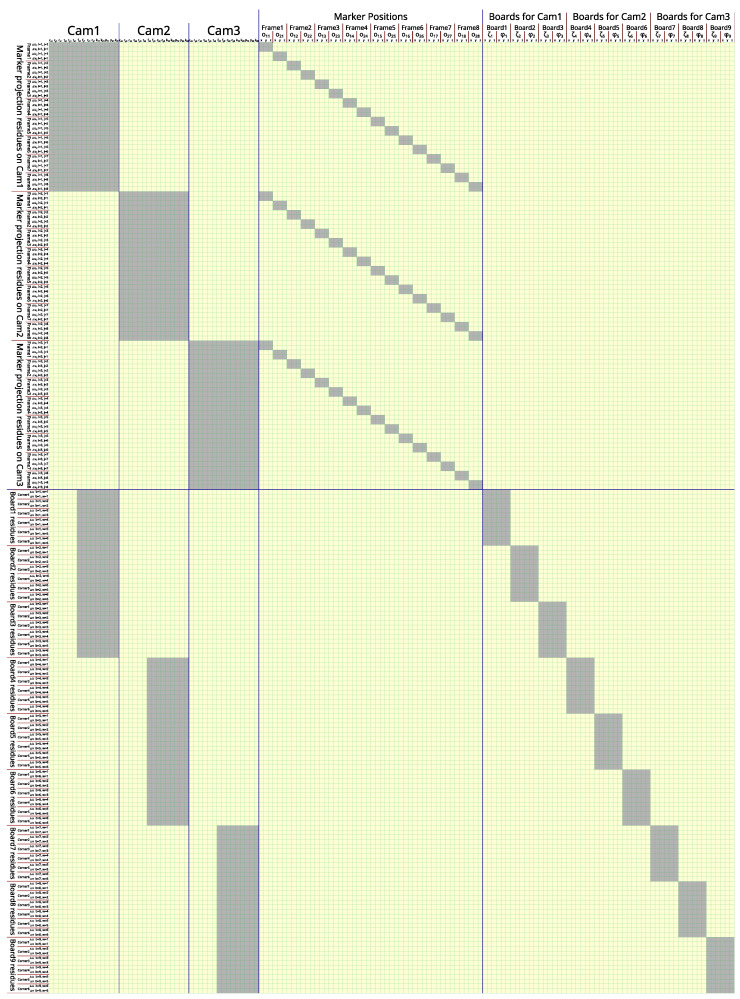
Structure of a Jacobian matrix *J* when N=3,M=8,B=9,W=6, and each camera has 3 chessboard images. Each column represents an unknown parameter in $\vec{\theta }$, and each row represents a residue in the residue vector ($\vec{e}$). The cells that are not painted with gray are always zero. This sparse structure allows for a huge acceleration in our customized forward-mode automatic differentiation. Zoom for more details in the digital version.

**Figure 7 sensors-24-07416-f007:**
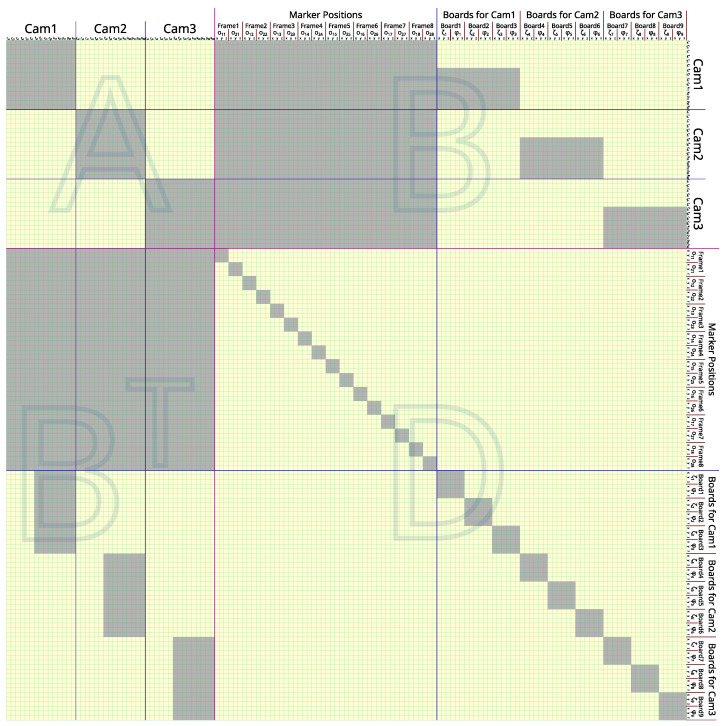
Structure of $J^{⊤}J+\lambda I$ when N=3,M=8,B=9,W=6, and each camera has 3 chessboard images. The cells that are not painted with gray are always zero. This matrix is split into matrices $A,B,B^{⊤}$, and D according to Equation (Equation 8). Zoom for more details in the digital version.

**Figure 8 sensors-24-07416-f008:**
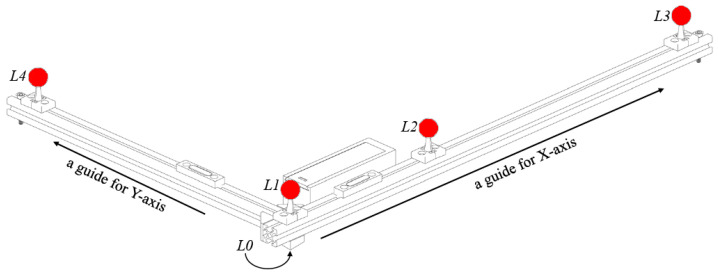
The optional L frame with 4 active red markers. *L*0 is the guide for the origin.

**Figure 9 sensors-24-07416-f009:**
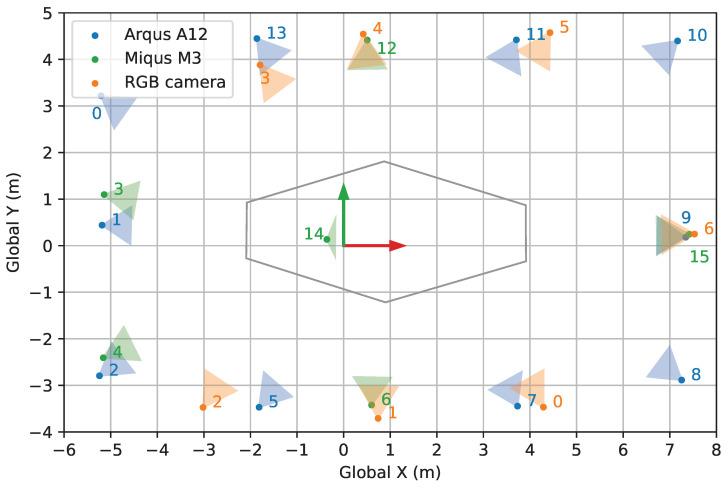
Top view of the experimental setup showing the spatial arrangement of marker-based cameras (Arqus M12 and Miqus M3), RGB cameras, and the hexagonal target capture area in relation to the global coordinates.

**Figure 10 sensors-24-07416-f010:**
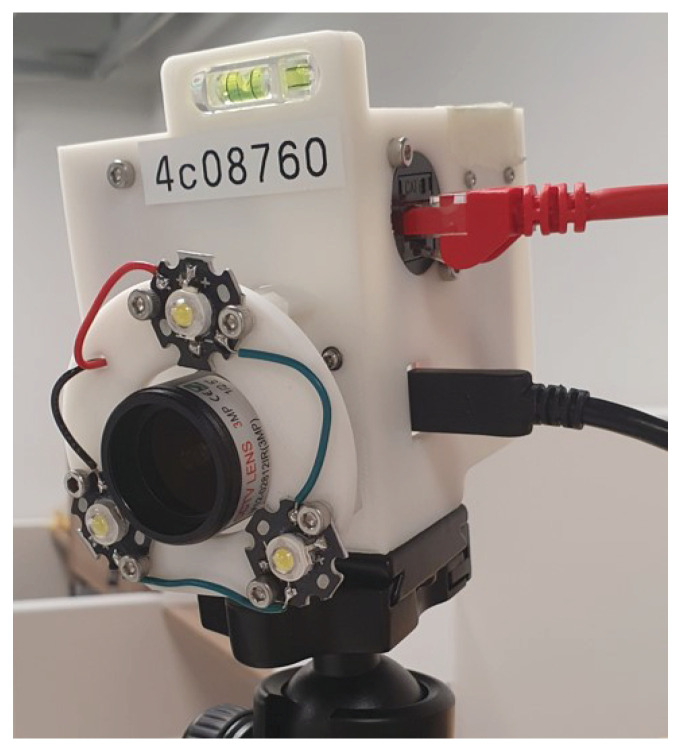
An RGB camera with a varifocal lens. The three LEDs are added to support mocap-assisted calibration and the benchmarking record.

**Figure 11 sensors-24-07416-f011:**
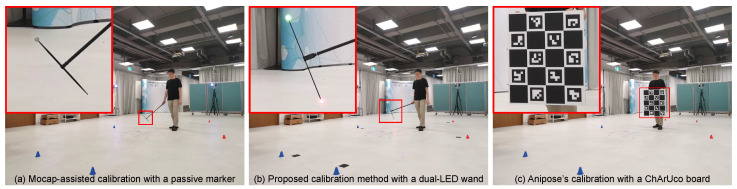
Three different modes of calibration.

**Figure 12 sensors-24-07416-f012:**
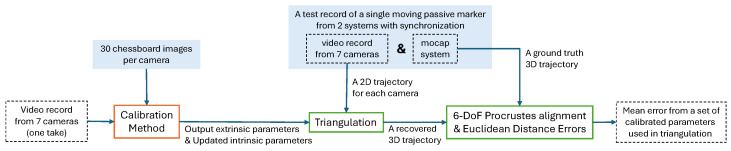
The proposed benchmarking workflow using the error between the recovered 3D trajectory in comparison with an independent ground-truth record. This benchmarking workflow is repeated 33 times with 33 different takes of video record input (i.e., 33 takes of wand waving or 33 takes of ChArUco board waving) to test the consistency of the method. Note that the data in the blue box are kept constant to ensure fairness across different calibration methods. In particular, for mocap-assisted calibration, the method also has access to the 3D trajectory of the passive marker for the corresponding take, which is not included in this figure.

**Figure 13 sensors-24-07416-f013:**
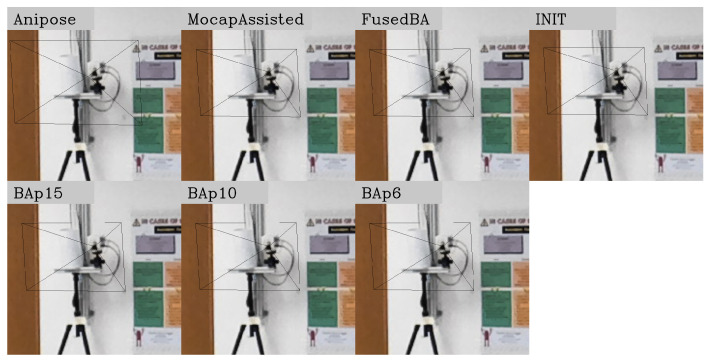
Projections of a camera’s position and frustum from an opposite camera’s perspective using parameters from different calibration methods. The result from Anipose’s calibration produces the worst alignment, as the frustum center is off relative to the camera lens. INIT also exhibits noticeable misalignment, while the rest of the methods align well.

**Figure 14 sensors-24-07416-f014:**
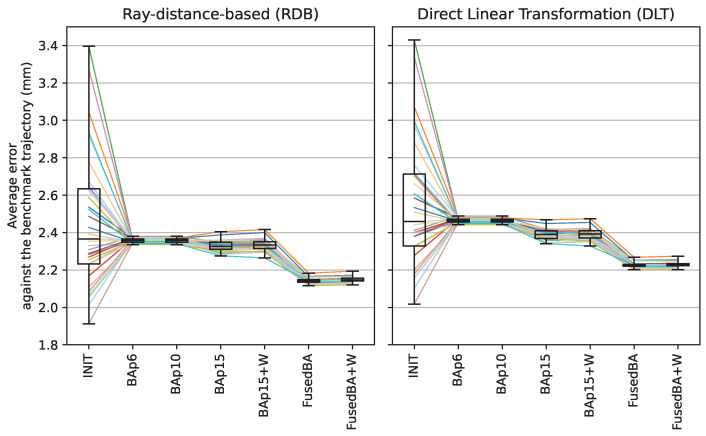
Distribution of benchmark errors from 33 rounds of calibration across different variants of bundle adjustment and the two triangulation methods.

**Figure 15 sensors-24-07416-f015:**
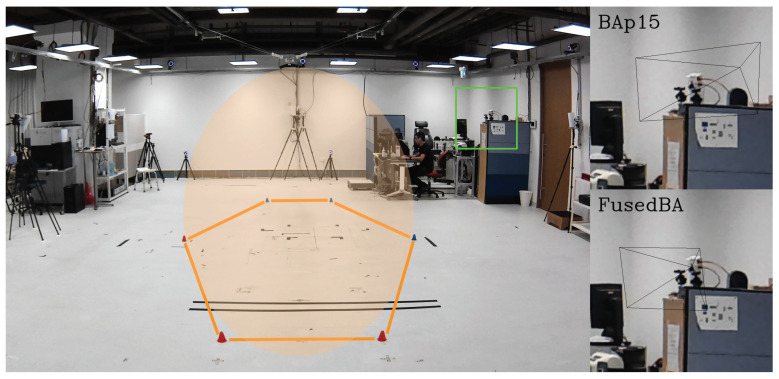
From some camera perspectives, the area of wand waving in the capture volume cannot cover the majority of the FOV. In this example, the wand is only waved in the highlighted area in the middle. As a result, the intrinsic parameters fine-tuned by the BAp15 method become overfitted to the wand data and cause the projection outside the wand-waving area to be badly distorted. However, the proposed FusedBA method can still fine tune all the intrinsic parameters without the mentioned issue.

**Table 1 sensors-24-07416-t001:** Statistical analysis of average errors from 33 rounds of each calibration method.

	Errors on Benchmark Record (mm)		Wand RMS Projection Error
Method	RDB	DLT	After Optimization (px)
Anipose	10.05912 ± 2.07237	10.67667 ± 2.77518	N/A
MocapAssisted	1.35763 ± 0.00705	1.42898 ± 0.00808	N/A
INIT	2.46065 ± 0.35063	2.54795 ± 0.33378	0.84400 ± 0.13966
BAp6	2.35667 ± 0.01020	2.46403 ± 0.01028	0.31793 ± 0.00447
BAp10	2.35667 ± 0.01020	2.46403 ± 0.01028	0.31793 ± 0.00447
BAp15	2.32816 ± 0.02848	2.39151 ± 0.02804	0.23108 ± 0.00419
BAp15+W	2.33387 ± 0.03047	2.39213 ± 0.02938	0.23390 ± 0.00418
FusedBA	2.14310 ± 0.01418	2.22776 ± 0.01427	0.26048 ± 0.00365
FusedBA+W	2.14911 ± 0.01470	2.22979 ± 0.01464	0.26230 ± 0.00367

**Table 2 sensors-24-07416-t002:** Statistical analysis of scaling factors from 33 rounds of each calibration method.

	Scaling Factor (Mean ± SD)	
Method	RDB	DLT
Anipose	0.995139 ± 0.001593	0.994237 ± 0.002577
Mocap-assisted	1.000011 ± 0.000023	1.000022 ± 0.000025
INIT	0.999571 ± 0.000338	0.999601 ± 0.000328
BAp6	1.000007 ± 0.000036	1.000018 ± 0.000026
BAp10	1.000007 ± 0.000036	1.000018 ± 0.000026
BAp15	0.999804 ± 0.000119	0.999845 ± 0.000130
BAp15+W	0.999786 ± 0.000103	0.999821 ± 0.000117
FusedBA	0.999967 ± 0.000071	0.999994 ± 0.000033
FusedBA+W	0.999944 ± 0.000085	0.999972 ± 0.000065

**Table 3 sensors-24-07416-t003:** Results from paired t-tests with an alternative hypothesis of fewer errors from RDB triangulation than from DLT triangulation (N=33).

	Paired t-Test Results	
Method	t-Statistic	*p*-Value
Anipose	−4.11	$1.38\times 10^{-4}$
Mocap-assisted	−160.84	$2.06\times 10^{-48}$
INIT	−16.41	$1.93\times 10^{-17}$
BAp6	−576.97	$3.71\times 10^{-66}$
BAp10	−576.96	$3.71\times 10^{-66}$
BAp15	−88.25	$4.33\times 10^{-40}$
BAp15+W	−96.75	$2.31\times 10^{-41}$
FusedBA	−222.79	$6.18\times 10^{-53}$
FusedBA+W	−214.40	$2.11\times 10^{-52}$

## Data Availability

The data recorded in this study are made available for research purposes at koonyook.github.io/MCalib with access code b875UZ (accessed on 15 November 2024).

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
