# Peer review of "Multi-Camera Calibration Using Far-Range Dual-LED Wand and Near-Range Chessboard Fused in Bundle Adjustment"

_sensors, 2024, doi:10.3390/s24237416_

Round 1

Reviewer 1 Report

Comments and Suggestions for Authors

Authors presents a multi-camera calibration approach using far-range dual-LED wand and near-range chessboard fused in bundle adjustment. This method utilizes the chessboard information from the intrinsic calibration step to help constrain the intrinsic parameters throughout the camera field of view, including areas beyond the reach of wand waving. Although the proposed idea seems reasonable and the experimental results look good, there are a few key issues that need to be addressed carefully.

1.     Section 3 describes the details of the proposed calibration method and the design of tools. The organization structure should be arranged according to the working procedure, Intrinsic Parameter Initialization-> Extrinsic Calibration with a Dual-LED Wand. Should Section 3.1 (Ray-Distance-Based (RDB) Triangulation) be included in Subsection 3.3.6?

2.     The principles description of proposed method is insufficient. For example, how the 2D centers of all visible red and green is detected? There are numerous image processing and detection methods of well recognized quality and robustness, including gray center method, ellipse detection method, template matching method, which should be selected as competitive solutions. Another key problem is the definition of global coordinate system. The information given in Section 3 is quite ambiguous to the ones who want to follow this work.

3.     The authors provided details of the principles and models which should combine schematic diagram to enhance understanding.

4.     In Figure 3, how is visualized full-field result graph obtained by concatenating different camera views?

5.     What is the difference between the ray-distance-based objective function in Section 3.1 and the classical physical optimization function?

6.     In subsection 3.3.5, authors extract the extrinsic matrix from the essential matrix obtained according to epipolar geometry constraint. Does this require multiple synchronized global-shutter RGB cameras surrounding a large capture volume have a common FOV to gather a list of all 2D-2D correspondence pairs?

7.     In Section 3.3.7, the scale factor is adjusted based on the actual size of the wand length. However, since the checkerboard also contains real scale information during the bundle adjustment (BA) process, will the adjustment of the scale factor lead to a scaling of the distances between the corners of the checkerboard?

8.     Lines 466-467 only mention that the camera's horizontal distance is between 3.7m and 6.6m. It is necessary to further clarify the size information of the checkerboard, the shooting distance of the checkerboard, and the camera's depth of field. When capturing the checkerboard, can the camera ensure that both the checkerboard and the wand are free from defocus?

9.     In Section 5, the results obtained by the proposed calibration method using RDB triangulation can be compared with those obtained by the traditional bundle adjustment optimization method minimizing the sum of reprojection error.

10.  Authors could obtain an accurate measurement of the distance between the centers of the two markers on the wand using close -range photogrammetry, and then add wand-length constraint in the bundle adjustment.

Comments on the Quality of English Language
The authors should make a proof-reading of the entire manuscript and improve English expression.

Reviewer 2 Report

Comments and Suggestions for Authors

The manuscript presents a novel multi-camera calibration method using a dual-LED wand and chessboard fusion within bundle adjustment.

The methodology is detailed, and the benchmarking workflow is adequate. After reading the manuscript, I believe it fits the journal well.

There is a minor flaw I noticed in the methodology section that could be improved. I suggest discussing alternative solutions and explaining why your method is sufficient.

In section 3.3.7, it seems the authors did not constrain the distance between two markers in the bundle adjustment, though there are many standard solutions for this.

For example, you can add a distance constraint between two markers:

argmin (|| o1i - o2i || - intended_wand_length )^2

This constraint should be applied for each frame.

For your reference, a 6DoF constraint looks like the one in the following paper, but you only need a translation constraint. Please refer to the Stereo Constraint section, Equation 2 in:

Zhang, Shuo, et al. "Self-Calibration of the Stereo Vision System on the Chang’E-5 Probe Based on Images and Robot Arm Footprints." Photogrammetric Engineering & Remote Sensing 89.11 (2023): 713-719.

Another solution assumes the wand is a perfectly rigid body:

Parameterize o2 with o1, for example, o2 = intended_wand_length \* \[cos(theta)*sin(phi), cos(theta)*cos(phi), sin(theta)], where [theta, phi] are the direction of \arrow{o1o2}. This approach reduces the o1o2 6DoF to 5DoF and naturally constrains the wand length.

In addition, the chessboard length could be used for scale calibration as well.
